# The Impact of Cognitive Load on Cooperation and Antisocial Punishment: Insights from a Public Goods Game Experiment

**DOI:** 10.3390/bs14080638

**Published:** 2024-07-25

**Authors:** Yanru Zhao, Zhuoran Li, Shan Jin, Xiaomeng Zhang

**Affiliations:** 1College of Economics and Management, Northwest A&F University, Xianyang 712100, China; 15315912325@nwafu.edu.cn (Y.Z.);; 2Economics Experimental Lab, Nanjing Audit University, Nanjing 210017, China; jinshan@nau.edu.cn

**Keywords:** cognitive load, PGG, antisocial punishment, C91, D91

## Abstract

This paper investigates the impact of cognitive load on the formation and maintenance of cooperation within a public goods game experiment featuring a punishment option. By integrating the experimental designs of prior studies and manipulating cognitive load through the memorization of numbers with varying digits, we reveal that high cognitive load accelerates the breakdown of cooperation, irrespective of the presence of a punishment system. Furthermore, under high cognitive load, participants are more likely to engage in antisocial punishment, while the punishment of free riders remains unaffected. These findings suggest that increased cognitive load depletes the cognitive resources needed for deliberative decision-making, leading to a higher propensity for antisocial punishment. Our study contributes to the literature by demonstrating the significant influence of cognitive load on cooperative behavior and providing new insights into the causes of antisocial punishment.

## 1. Introduction

Cooperation is the most fundamental basis for the development of human society. (On the occasion of its 125th anniversary, the journal Science listed the issue of cooperation as one of the 125 most significant scientific questions for humanity [1].) Understanding how to achieve and sustain human cooperation effectively is paramount for the development and prosperity of human societies [2,3,4,5]. Previous studies have explored the design of mechanisms to promote and sustain cooperation from various perspectives, including social preferences [6,7,8], reciprocity systems [9,10,11], reward systems [12,13,14,15], repeated games [16,17,18], conditional cooperation [19,20,21,22], and homophily [23,24]. For example, Trivers [6] demonstrated that humans retain altruistic preferences through evolution (altruistic preferences are not eliminated by natural selection), which, in turn, maintains human cooperation. Fehr and Fischbacher [7] showed that altruism and cooperation are inherent to human nature. Fischbacher and Gächter [8] illustrated that, although the willingness to cooperate declines as the experiment progresses, humans still possess the willingness and motivation to cooperate. Nowak and Sigmund [9,10] demonstrated that if reciprocity mechanisms exist in society, they can effectively ensure human cooperation. Sigmund et al. [13] found that providing additional rewards to participants can effectively promote cooperation. Yang et al. [15] used endogenous rewards, such as collecting taxes from everyone and allowing participants to reward their partners, which also promoted cooperation. Fudenberg and Maskin [17] showed that, when cooperation is repeated, participants choose to cooperate to achieve greater future gains. Charness and Yang [24] demonstrated that sometimes participants’ decision to cooperate depends on whether others are cooperating, indicating that cooperation is conditional. Furthermore, Fischbacher et al. [20] showed that, in this behavior pattern, if participants are allowed to choose their partners, those inclined to cooperate tend to gather together.

Among the various mechanisms, the introduction of punishment options to penalize free riders has been shown by numerous studies to effectively enhance the efficiency of cooperation and maintain its stability [25,26,27,28,29,30,31]. However, numerous studies have demonstrated that the introduction of punishment also leads to the emergence of antisocial punishment. Antisocial punishment refers to the act of punishing a cooperative individual rather than a free rider. This form of punishment is clearly detrimental to cooperation and results in unnecessary social welfare losses [32,33,34,35,36]. While antisocial punishment is frequently observed in research, the reasons behind participants’ choice to engage in such behavior remain an area that requires further investigation.

Undoubtedly, both the choice to cooperate and the decision to punish others, along with whether such a punishment contributes to sustaining cooperation, require further consideration. This inevitably involves the concept of bounded rationality [37], questioning whether individuals’ cognitive capacities can bear the cognitive load necessary for these decisions. Previous research has demonstrated that cognitive load significantly impacts individuals’ risk decisions [38,39,40,41,42,43,44], intertemporal choices (time preferences) [43,45], trust [46], and economic rationality [43,47,48]. For example, studies have indicated that individuals under higher cognitive load tend to be less patient and more risk-averse [43] (however, see Ball et al. [49], who did not replicate the results of Deck and Jahedi [43]). Ref. [46] demonstrates that, as cognitive load increases, individuals’ levels of trust decrease. However, research on the impact of cognitive load on the formation and maintenance of cooperation is currently lacking. This paper aims to explore the effects of cognitive load on the establishment and sustainability of cooperation, as well as on the use of punishment mechanisms, in an effort to fill this gap in the academic literature.

We are concerned with this issue for several reasons. Firstly, prominent economists have theoretically demonstrated that altruistic behavior is closely related to self-control [50]. There is also a substantial body of psychological literature showing that self-control requires high cognitive ability and cognitive load [51,52,53]. Additionally, other psychologists and neuroscientists have proven that human prosocial behavior (cooperation) and the efforts to maintain such behavior require self-control and deep thinking [54,55,56]. This leads to the hypotheses of this paper: (1) Since prosocial behavior requires high self-control, cognitive load could reduce self-control, thereby decreasing prosocial behavior. (2) If the effort to maintain prosocial behavior requires self-control and careful consideration, cognitive load could reduce the reliability of this effort (e.g., leading to the misuse of punishment or antisocial punishment).

In this study, we integrated the experimental designs of Deck and Jahedi [43] and Fehr and Fischbacher [7], Fehr and Gächter [25] to investigate the effects of cognitive load on cooperation in a public goods game experiment with a punishment option. By manipulating participants’ cognitive load through the memorization of numbers with varying digits, we found that, under high cognitive load, the rate of cooperation breakdown accelerates regardless of the presence of a punishment mechanism. Another interesting finding is that the proportion of participants using punishment significantly increases under high cognitive load, driven by an increase in antisocial punishment, whereas punishment of free riders is not affected by cognitive load. We argue that increased cognitive load depletes the cognitive resources available for considering whether to punish, thereby making participants more inclined to engage in antisocial punishment.

Our research makes the following contributions to the existing literature. First, we reveal the impact of cognitive load on cooperation, finding that cognitive load leads to the rapid breakdown of cooperation. This expands the literature on the effects of cognitive load on economic decision-making [38,39,40,41,42,43,45,46,49]. In particular, Døssing et al. [57] effectively explored the impact of cognitive load (time pressure) on cooperation and found that high cognitive load increases initial willingness to cooperate, whereas we employed an entirely exogenous cognitive load. Our aim was to simulate real-life cognitive loads that do not arise from the task itself but from the environment in which the decision-maker is situated, such as the increased cognitive load due to poverty or hunger. Second, we partially uncover the reasons behind the use of antisocial punishment. We find that, when individuals face high cognitive costs and cannot sufficiently consider the costs and benefits of punishment, they tend to engage in antisocial punishment. This provides new insights into the causes of antisocial punishment [58,59,60].

The rest of this paper is organized as follows. Section 2 introduces our experimental design and procedure. Section 3 presents our experimental results. In Section 4, we will provide explanations and discussions on various issues related to our experimental design and parameter choices. Finally, Section 5 offers our conclusions.

## 2. Experiment

### 2.1. Design

In the present study, we designed a 2×2 experimental setup with two dimensions: cognitive load (high vs. low) and the presence of punishment (with vs. without; see Table 1). In the high cognitive load treatment, participants were required to memorize an eight-digit random number, whereas in the low cognitive load treatment, participants only needed to memorize a three-digit random number. Participants who could correctly recall the full number at the end of the experiment received an additional and 20 as a reward.

In the non-punishment treatment, we utilized a linear social dilemma game without punishment opportunities to measure cooperation [7,25]. Participants were randomly paired into two-person groups, and each participant received 10 tokens (each token was equivalent to RMB 2 Yuan). Participants decided how many tokens to contribute to a public goods pool. Each token contributed to the pool was increased by a factor of 1.6 and then equally distributed between the two participants. The payoff function for participant *i* in the first stage is described by Equation (Equation 1):(1)$$\pi_{i}^{1}=10-g_{i}+1.6*\left(g_{i}+g_{j}\right)/2$$
where $g_{i}$ represents the tokens contributed to the public goods by participant *i*, and $g_{j}$ represents the tokens contributed by participant *j*.

In the punishment treatment, participants had the option to allocate tokens to punish their counterpart after observing their own payoff $\pi_{i}^{1}$. They could choose to impose a punishment at a cost of 1 to 3 tokens (Participants could only choose whole numbers, i.e., 1, 2, or 3). Each token used for punishment inflicted a threefold loss on the other participant. The final payoff for participant *i* is represented by Equation (Equation 2):(2)$$\pi_{i}^{final}=\pi_{i}^{1}-p_{ij}-3p_{ji}$$
where $p_{ij}$ denotes the number of punishment tokens assigned by participant *i* to participant *j*, and $p_{ji}$ denotes the number of punishment tokens assigned by participant *j* to participant *i*.

To differentiate between punitive actions targeting free riders and those aimed at enhancing cooperation, we defined antisocial punishment as per [58,61], which involves punishing a participant who contributed an equal or greater amount than the punisher themselves. The experimental procedure was repeated for 10 rounds, with participants being randomly re-paired after each round.

The details of our treatments are summarized in Table 1.

### 2.2. Procedures and Sample

We recruited 256 students (theselection of the number of participants was based on a pilot experiment we conducted prior to the main study, combined with a power analysi; we ultimately decided that 60 participants per treatment group was an appropriate number) from the subject pool at the Experimental Economics Lab at Nanjing Audit University (thesubject pool consists of approximately 2000 students at Nanjing Audit University, including both undergraduate and graduate students) to participate in the experiment. These students were randomly assigned to one of the four treatment conditions.

The experimental process comprised five stages: sign-in, memorization of a random number (thememorization stage lasted for 1 min and could not be skipped), a 10-round experiment, number recall, and a follow-up survey.

During the sign-in procedure, participants were asked to read and sign a consent form that had been approved by the Nanjing Audit University Institutional Review Board. In the memorization stage, participants were given either an 8-digit random number (high cognitive load) or a 3-digit random number (low cognitive load) to memorize. Participants will be informed that they will need to recall and write down these numbers at the end of the experiment. Those who correctly write down all the numbers will receive a reward of RMB 20. After the memorization phase, participants will be randomly paired with another participant to engage in a public goods game. In the no-punishment treatment, participants will be informed of the rules of the public goods game and that their partner will change each round. They will also be told that one of their ten decisions will be randomly selected for payment. In the punishment group, participants were informed that, after each round of decision-making, they would have the option to punish their game partner, and the punishment rules were explained to them. Similarly, one of their rounds would be randomly selected for payment. Each round of the experiment was conducted on computers, and participants were instructed to refrain from speaking to one another throughout the process. This experiment was conducted using oTree [62]. The experiment was divided into 8 sessions, each with 30 to 36 participants. All sessions were completed within 2 days, with 4 sessions conducted each day (for more details, please refer to Appendix A). Upon completing all rounds of the experiment, participants were asked to recall the number they had memorized. Subsequently, a demographic survey was administered to collect essential personal information. Upon completion of all these procedures, participants were compensated privately and then dismissed.

We reported the demographic summary statistics in the left panel of Table 2. We also tested the differences in the demographics of the subjects between four treatments and reported them in the right panel. Among the four treatments, there are no significant differences in the basic demographic information of subjects, so we conclude that there is no significant selection bias in any treatment, which is in line with the principle of random selection. In addition, the age of the subjects in each wave ranges from 19 to 23 years old, nearly 74% of the subjects are women, and approximately 29% are majoring in economics, which is consistent with the student structure of Nanjing Audit University.

## 3. Results

In this section, we will first present the impact of cognitive load on cooperative behavior, followed by a detailed analysis of its influence on punishment choices in the presence of a punishment mechanism.

### 3.1. Cooperation Behavior

The results obtained for cooperative behavior are shown in Figure 1. We found that, in the initial stages, there were no significant differences in the contribution levels across the various treatments. However, as the experiment progressed, contribution levels began to decline. The treatment group with high cognitive load and no punishment option experienced the fastest decline, with cooperation significantly collapsing by the 10th round. Both the high cognitive load with punishment and low cognitive load without punishment treatments showed a downward trend in cooperation, but not as rapidly as the high cognitive load without punishment treatment. There were no significant differences between these two treatments. The group with low cognitive load and a punishment option maintained the highest level of cooperation.

These results confirm previous findings that the presence of a punishment option is more effective in maintaining cooperation than treatments without punishment [7,25], regardless of the cognitive load. Additionally, we found that cognitive load leads to a significantly faster rate of cooperation collapse, irrespective of the presence of a punishment option. This outcome remains robust across different conditions.

We next turn to the regression results reported in Table 3, which allow us to examine the effect of cognitive load and a punishment option on cooperation willingness and its rate of decline. Specifically, we adopt the following linear regression specification:(3)$$Y_{it}=\beta_{1}\times {\mathrm{HCL}}_{i}+\beta_{2}\times {\mathrm{Punish}}_{i}+\beta_{3}\times {\mathrm{HCL}}_{i}\times {\mathrm{Punish}}_{i}+\beta_{4}\times t+X_{i}^{\prime }\gamma +e_{i},$$
where $Y_{it}$ is subject *i*’s contribution to the public pool in round *t*, and ${HCL}_{i}$ is a variable that indicates the high cognitive load treatment, which is a dummy variable. ${HCL}_{i}$ equals 1 when participants are in the high cognitive load treatment; otherwise, ${HCL}_{i}$ equals 0. ${Punish}_{i}$ is a variable that indicates the treatment with punishment; when the punishment option is available, its value is 1; otherwise, it is 0. Our regression analysis also includes interaction terms, implying that we use the low cognitive load and no punishment treatment group as the reference group. $X_{i}$ is a vector of control variables that includes age, gender, family income, and major, and $e_{i}$ is an error term clustered at the subject level. $\beta_{1}$ measures the effect of cognitive load only on cooperation, $\beta_{2}$ measures the effect of the punishment option only on cooperation, and $\beta_{3}$ measures the combined effect of cognitive load and the punishment option. We report the Ordinary Least Squares (OLS) results in columns (1) and (2). To further assess the impact of cognitive load and the punishment option on the speed of cooperation collapse, we included interaction terms between each of the three variables and the round number *t* in our regression analysis. The results are presented in columns (3) and (4).

From columns (1) and (2), we can see that an increase in cognitive load significantly reduces the average contributions to the public goods pool, indicating a decrease in the average willingness to cooperate. Conversely, the presence of a punishment option increases the willingness to cooperate. For all treatments, the willingness to cooperate declines as the number of rounds increases. Columns (3) and (4) show that both cognitive load and the punishment option significantly affect the rate of cooperation decline. High cognitive load accelerates the collapse of cooperation, while the presence of a punishment option somewhat mitigates this rate of decline. Our interaction terms indicate that, as the experiment progresses, the impact of cognitive load on the breakdown of cooperation becomes increasingly significant. This means that, the longer the memory task, the less willing participants are to contribute to public goods. Conversely, as the experiment continues, the punishment system becomes increasingly effective in maintaining cooperation.

### 3.2. Punishment Behavior

We divided punishment behavior into two types. The first type consists of participants who punish a partner who has contributed less than themselves, which is defined as the punishment for free-riding behavior. The second type consists of participants who punish a partner who has contributed the same or more than themselves, which is defined as antisocial punishment.

We calculated the average number of rounds and the average number of tokens used for punishment in each treatment (Figure 2a,d). Additionally, we measured the rounds and tokens used for antisocial punishment (Figure 2b,e), as well as for punishing free riders (Figure 2c,f) (theaverage number of punishment tokens used, as we mentioned, refers to the average over the 10 rounds).

Figure 2a,d illustrate that, with an increase in cognitive load, participants are significantly more inclined to punish others (average punishment rounds: HCLR=4.516 vs. LCLR=2.515) (HCLR and LCLR represent the punishment rounds in high cognitive load treatment and low cognitive load treatment, respectively) and are willing to use more tokens for punishment (average punishment tokens per round: HCLT=0.919 vs. LCLT=0.495) (HCLT and LCLT represent the punishment tokens in high cognitive load treatment and low cognitive load treatment, respectively). These differences are statistically significant (for average punishment rounds: *p*-value < 0.001; for average punishment tokens: *p*-value < 0.001).

When breaking down these punishment results into antisocial punishment (Figure 2b,e) and punishment for free riders (Figure 2c,f), we observe that the punishment for free riders does not significantly increase with cognitive load (for average punishment rounds: HCLR=1.891 vs. LCLR=1.545, *p*-value =0.099; for average punishment tokens: HCLT=0.389 vs. LCLT=0.306, *p*-value =0.089). (Based on our anonymous reviewer’s suggestion, we examined the accuracy of punishing free riders, specifically the proportion of free riders who were punished. The results show that the punishment rate for free riders is slightly higher in the low cognitive load group compared to the high cognitive load group, but the difference is not significant (HCL: 73% vs. LCL: 76%, *p*-value = 0.451).) However, antisocial punishment significantly increases with cognitive load (for average punishment rounds: HCLR=2.625 vs. LCLR=0.970, *p*-value <0.001; for average punishment tokens: HCLT=0.530 vs. LCLT=0.189, *p*-value <0.001).

These findings indicate that, as cognitive load increases, the punishment behavior of participants significantly rises, primarily driven by an increase in antisocial punishment.

We next turn to the regression results reported in Table 4, which allow us to examine the effect of cognitive load on punishment behavior, especially antisocial punishment behavior. In the left panel of Table 4, we report the estimates from a probit model in Equation (Equation 4):(4)$$P_{it}=\beta \times {\mathrm{HCL}}_{i}+X_{i}^{\prime }\gamma +e_{i},$$

In the right panel of Table 4, we report the estimates from an ordered logit model in Equation (Equation 5):(5)$${\mathrm{Token}}_{it}=\beta \times {\mathrm{HCL}}_{i}+X_{i}^{\prime }\gamma +e_{i},$$
where $P_{it}$ is equal to 1 if subject *i* chooses to punish their partner in round *t*, $Token_{it}$ is the number of tokens that subject *i* uses to punish their partner in round *t*, ${\mathrm{HCL}}_{i}$ is a variable that indicates the high cognitive load condition, $X_{i}$ is a vector of control variables that includes round numbers, age, gender, income level and major, and $e_{i}$ is an error term clustered at the subject level.

We reported the results of total punishment in columns (1) and (4), the punishment for free riders in columns (2) and (5), and the antisocial punishment in columns (3) and (6). From Table 4, we can conclude that, as cognitive load increases, the punishment behavior and the tokens used for punishment increase significantly. Among all punishments, those for free riders increase slightly, while the antisocial punishment increases very significantly.

Another noteworthy point is that we analyzed the success rates of number recall in both the high cognitive load group and the low cognitive load group. Firstly, we found that the low cognitive load group almost entirely recalled the numbers accurately, with a perfect recall rate of 98.46%. The high cognitive load group had a relatively lower probability of perfect recall at 87.30%, with a 94.44% probability of recalling more than three numbers. This demonstrates, firstly, that most participants in the high cognitive load group indeed bore a higher cognitive load by remembering more than three numbers. Secondly, it shows that the rate of recalling more than three numbers starts to decrease, indicating an effective manipulation of cognitive load.

Regarding the use of punishment tokens among participants in the high cognitive load group who remembered more and fewer numbers, we found that those in the high cognitive load group who recalled three or fewer numbers indeed used fewer punishment tokens. The difference between this subgroup and the low cognitive load group was not significant (p=0.146), but there was a significant difference between this subgroup and those who remembered more than three numbers (p=0.032). This indicates that, when participants reduce their cognitive load by not remembering more numbers, their punitive behavior also decreases accordingly.

## 4. Discussion

In this study, we opted to use an exogenous cognitive load, such as remembering a few numbers, rather than an endogenous one, such as the difficulty of the experiment itself. The reason for focusing on an exogenous cognitive load is that this paper aims to simulate and describe scenarios in which individuals face cooperation dilemmas and public goods problems under varying cognitive abilities or environments. For instance, as highlighted by Banerjee and Duflo [63], individuals living in poverty experience greater cognitive load when making decisions (due to a lack of sufficient energy), which inevitably affects the quality of their decisions. We aim to address this issue. Of course, as our reviewers have pointed out, the cognitive load associated with increased decision-making difficulty is also worth considering. We hope that future research will explore this issue further.

The reason we chose not to have participants memorize a new number each round, but instead to remember a single number throughout the experiment, is twofold. Firstly, each round is relatively short (approximately 30 s). If participants were only required to remember a number for 30 s, it would be difficult to distinguish between high and low cognitive load (i.e., the cognitive resources consumed in remembering an eight-digit number for 30 s is not significantly different from that for a three-digit number; longer durations would amplify the cognitive resource consumption of longer numbers). Secondly, having participants frequently memorize multiple different numbers would also increase the cognitive load for the low cognitive load group, making it challenging to define it as a truly low cognitive load.

For the multiplier of 1.6 used in the public goods game, we based our choice on Fehr and Gächter [25,27]. They found that 1.6 is an effective number to capture the characteristics of public goods. Specifically, it ensures that the marginal benefit to the individual is slightly lower than that of private goods (0.8 vs. 1). However, this reduction is acceptable and does not create an absolute disadvantage that would completely discourage individuals from contributing to public goods. Through repeated experiments, they determined that 1.6 is an appropriate number. Regarding why we chose a 3× multiplier for the punishment, this is also based on Fehr and Gächter [25]’s experimental research. They found that a multiplier of three is appropriate because it makes the punishment effective, ensuring that the punished party incurs a loss. At the same time, it is not so large as to cause the entire experiment to collapse.

The reason we chose a random re-matching mechanism rather than using the same game partner throughout is to separate the reciprocity mechanism [25] from the reputation mechanism [64]. In games with the same opponent, players might engage in reciprocal punishment. For example, if A punishes B in one round, B might punish A in the next round as retaliation. If this happens, it becomes difficult to determine whether the increased retaliation in a specific treatment is due to cognitive load or if the players have randomly fallen into a cycle of retaliation. Additionally, the reputation mechanism also plays a role in cooperative games. When playing with the same partner repeatedly, participants might choose strategies based on their opponent’s reputation, which adds additional endogenous cognitive costs. To avoid these two issues, we chose the commonly used method of randomly re-matching participants with new game partners each round.

Regarding why we chose to reward only those who remembered all the numbers, rather than rewarding each correct number, as done by Cappelletti et al. [65], there are two reasons: (1) In our pilot data, we found that, if participants were rewarded for each correct number, many participants would give up trying to remember all the numbers or stop making efforts to remember more numbers, instead choosing to remember only three (similar to the low cognitive load group). This contradicts our intention to simulate exogenous cognitive load. (2) If we paid participants based on each correct number, it would create different endowments for different treatments, potentially introducing new endowment effect issues.

This paper could potentially connect future directions with the rich literature on social governance. For instance, Ostrom [66] explored how successful ‘commons’ and self-management of public goods often reduce the ‘costs’ associated with monitoring/supervision through (often creatively) structured collective activities and institutions. In our experiment and related previous experiments, players were informed of their partners’ contributions after each round. However, in the real world, gathering information about peers’ behavior usually incurs additional ‘costs’, which are often decisive factors in whether a community can self-manage, despite the possibility of free-riding. This endogenous ‘cognitive load’ could be a future research direction for this paper.

Additionally, some studies [67] indicate that many outcomes are entirely dependent on agents imitating peers with higher fitness and that agents apply the same behavior (whether ‘cooperation’ or ‘free-riding’) across multiple interactions/games they participate in. ‘Cognitive load’ can be a rationale for this hypothesis: agents behave the same across all interactions because they lack the cognitive capacity to simultaneously track many different games. This can be particularly interesting in certain scenarios, as this mechanism can actually promote/spread cooperative behavior in these networked public goods game models. In some cases, the inability to manage ‘cognitive load’ might lead to ‘positive’ effects.

## 5. Conclusions

This study delved into the significant impact of cognitive load on cooperative behavior and the propensity for antisocial punishment in a public goods game setting. By manipulating cognitive load through the memorization of numbers of varying lengths, we reveal that high cognitive load not only accelerates the breakdown of cooperation but also significantly increases the incidence of antisocial punishment. These findings underscore the crucial role of cognitive resources in sustaining cooperative behavior and the detrimental effects of cognitive overload on social welfare.

Our experiment demonstrates that, under high cognitive load, participants were more likely to engage in antisocial punishment—punishing cooperative individuals rather than free riders. This behavior can be attributed to the depletion of cognitive resources necessary for deliberative decision-making, leading to more impulsive and less socially beneficial actions. The increased cognitive burden impaired participants’ ability to evaluate the long-term benefits of cooperation and the social costs of antisocial punishment accurately. Consequently, the breakdown of cooperation was accelerated, highlighting the fragility of cooperative systems under cognitive strain.

Our findings align with previous research on the impact of cognitive load on decision-making. Studies such as those by Frederick [38] and Burks et al. [39] have shown that cognitive load affects economic preferences and strategic behavior. Specifically, Deck and Jahedi [43] demonstrated that cognitive load reduces patience and increases risk aversion. Our study extends these insights by showing that cognitive load not only influences individual decision-making but also affects cooperative dynamics in social settings.

Moreover, the increase in antisocial punishment under high cognitive load corroborates the findings of Herrmann et al. [58] and Nikiforakis [60] (in fact, this paper does not explore the issue of cultural differences. The literature we cited merely demonstrates that the willingness to engage in antisocial punishment varies across different cultural backgrounds. This could, of course, lead to an interesting future research topic: whether there are differences in responses to cognitive load on punishment across different cultural backgrounds), who highlighted the prevalence of antisocial punishment in various cultural contexts. By linking these behaviors to cognitive load, our study provides a deeper understanding of the cognitive underpinnings of antisocial punishment and its impact on cooperation.

Future research should explore interventions that can mitigate the negative effects of cognitive load on cooperation and punishment behavior. Investigating the role of cognitive training, environmental design, and policy adjustments can provide actionable insights into promoting sustainable cooperation. Additionally, examining the impact of different types of cognitive load, such as emotional or social load—of course, this also includes directly endogenous cognitive loads, such as the complexity of calculations—on cooperative behavior can further elucidate the complex interplay between cognitive resources and social dynamics.

## Figures and Tables

**Figure 1 behavsci-14-00638-f001:**
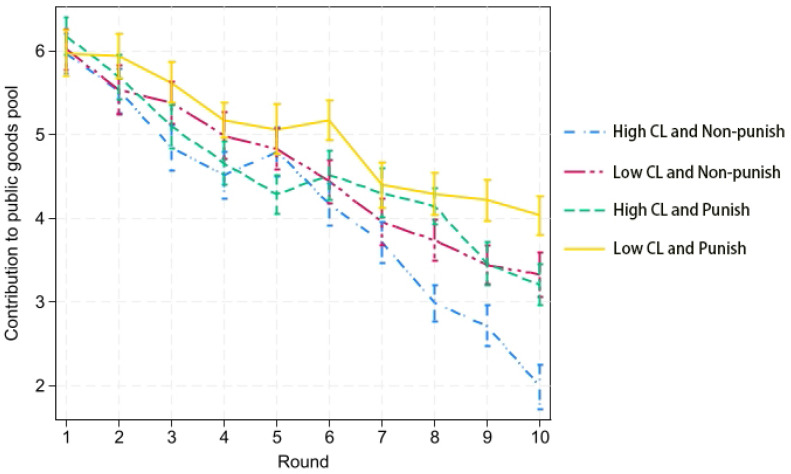
Contribution to the public goods pool in each round. Note: This figure depicts the contributions made to the public goods pool in each round in each treatment of the experiment. The *x*-axis represents the round number, while the *y*-axis represents the contribution tokens. The error bar denotes the 95% CI.

**Figure 2 behavsci-14-00638-f002:**
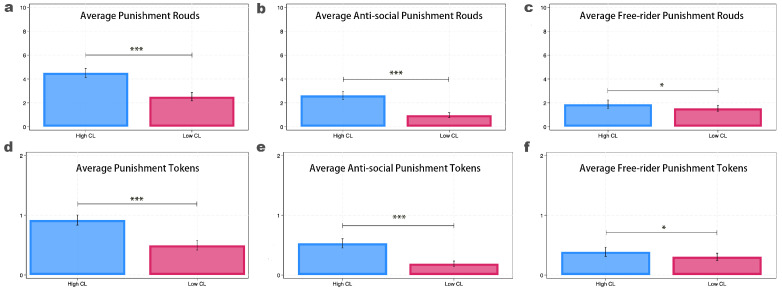
Punishment and antisocial punishment. Note: this figure shows the average number of rounds of punishment used by the subjects and the number of tokens used for punishment in each treatment. (**a**) The number of rounds in which participants used punishment. (**b**) The number of rounds in which participants engaged in antisocial punishment. (**c)** The number of rounds in which participants punished free riders. (**d**) The number of tokens used by participants for punishment. (**e**) The number of tokens used for antisocial punishment. (**f**) The number of tokens used to punish free riders. The error bar denotes the 95% CI. *** and * denote statistical significance at the 1% and 10% levels, respectively.

**Table 1 behavsci-14-00638-t001:** Treatment conditions.

	High Cognitive Load	Low Cognitive Load
Without Punishment	Memorize an 8-digit number	Memorize a 3-digit number
(ten rounds)	Standard PGG	Standard PGG
With punishment	Memorize an 8-digit number	Memorize a 3-digit number
(ten rounds)	PGG with punishment option	PGG with punishment option

**Table 2 behavsci-14-00638-t002:** Summary statistics and difference test.

	Summary Statistics				ANOVA
Variables	High CL and Non-Punish	Low CL and Non-Punish	High CL and Punish	Low CL and Punish	*p*-Values
	(1)	(2)	(3)	(4)	
Age	20.597	20.172	20.250	20.273	0.242
	(1.123)	(1.001)	(1.182)	(1.603)	
Female	66.1%	75.0%	82.8%	74.2%	0.202
	(47.7%)	(43.6%)	(38.0%)	(44.1%)	
Family Income	29.323	29.500	29.281	29.455	0.566
	(1.068)	(0.925)	(1.091)	(0.964)	
Econ Major	30.8%	30.9%	32.8%	34.5%	0.428
	(46.2%)	(46.3%)	(47.0%)	(47.6%)	
Observations	62	64	64	66	

Note: this table reports the demographic information for all subjects in each treatment. The left panel is the summary statistics; we report the means and the standard deviations in parentheses. The right panel is the ANOVA test of the difference among four treatments; we report the *p*-value.

**Table 3 behavsci-14-00638-t003:** Effect of cognitive load and punishment on cooperation.

	(1)	(2)	(3)	(4)
Variables	Contribution	Contribution	Contribution	Contribution
High cognitive load	−0.445 ***	−0.439 ***	0.132	0.135
	(0.057)	(0.057)	(0.124)	(0.124)
Punishment option	0.422 ***	0.426 ***	0.019	0.021
	(0.055)	(0.127)	(0.031)	(0.127)
High cognitive load × Punishment option	0.010	0.007	−0.248	−0.248
	(0.080)	(0.081)	(0.176)	(0.176)
Round × High cognitive load			−0.104 ***	−0.104 ***
			(0.020)	(0.130)
Round × Punishment option			0.073 ***	0.074 ***
			(0.020)	(0.019)
Round × High cognitive load × Punishment option			0.047	0.046
			(0.029)	(0.029)
Round	−0.309 ***	−0.309 ***	−0.306 ***	−0.307 ***
	(0.008)	(0.008)	(0.013)	(0.013)
Constants	6.261 ***	6.903 ***	6.247 ***	6.801 ***
	(0.060)	(0.643)	(0.084)	(0.654)
Demographics	No	Yes	No	Yes
Number of subjects	256	256	256	256
Number of observations	2560	2560	2560	2560
Adjusted $R^{2}$	0.442	0.442	0.459	0.459

Note: this table shows the effects of cognitive load and the presence of a punishment option on contributions to the public goods pool. Columns (1) and (2) present the impact on average contribution levels. Columns (3) and (4) display the impact on the trend of cooperation decline. Columns (1) and (3) do not control for demographic variables, while columns (2) and (4) control for participants’ age, gender, family income, and major. Robust standard errors clustered at subject level are reported in parentheses. ***, denote statistical significance at the 1% levels, respectively.

**Table 4 behavsci-14-00638-t004:** Effect of cognitive load on punishment behavior.

	Probability of Punishment Being Used			Tokens Used to Punish		
	(1)	(2)	(3)	(4)	(5)	(6)
Variables	Total	For Free Rider	Antisocial	Total	For Free Rider	Antisocial
High cognitive load	0.586 ***	0.019	0.844 ***	0.957 ***	0.039	1.532 ***
	(0.162)	(0.192)	(0.186)	(0.267)	(0.341)	(0.350)
Round	−0.001	−0.014	0.018	−0.001	−0.025	0.035
	(0.019)	(0.022)	(0.024)	(0.032)	(0.039)	(0.048)
High cognitive load × Round	−0.008	0.021	−0.034	−0.012	0.038	−0.062
	(0.026)	(0.030)	(0.031)	(0.043)	(0.054)	(0.057)
Constants	−1.280	−1.560	−1.587	−2.135	−2.705	−2.837
	(1.201)	(1.390)	(1.327)	(1.969)	(2.534)	(2.338)
Number of observations	1300	1300	1300	1300	1300	1300

Note: this table reports the estimation results for the effect of cognitive load on punishment behavior. All entries represent 130 subjects and 1300 observations. We controlled all demographics in Table 3. Columns (1), (2), and (3) report estimates of the probability that punishment is used, while Columns (4), (5), and (6) report estimates of the number of tokens used for punishment. Robust standard errors clustered at subject level are reported in parentheses. ***, denote statistical significance at the 1% levels, respectively.

## Data Availability

The data presented in this study are openly available in [Cognitive load and cooperation] at [DOI 10.17605/OSF.IO/XKGMN], reference number [XKGMN].

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
