# Peer review of "The Impact of Cognitive Load on Cooperation and Antisocial Punishment: Insights from a Public Goods Game Experiment"

_behavsci, 2024, doi:10.3390/bs14080638_

Round 1
Reviewer 1 Report
Comments and Suggestions for Authors
The article focuses on the role of individuals' limited capacities to handle cognitive loads as a factor that bounds rationality in decision making. This is widely understood to play a role when individuals make decisions while engaging in collective action (for example, whether to cooperate or free-ride in public goods games). More specifically, the article considers the interplay between (1) this bounded rationality and (2) individuals' decisions regarding punishment (that is, not just about whether or not to free-ride, but also whether or not to enact punishment). My impression is that this combination is the article's primary innovative contribution. "Antisocial punishment" is adopted here to describe the application of sanctions to cooperative peers who have been incorrectly identified as targets for punishment ("false positives" in the effort to detect free riders, we might say). The individual's own contribution to the public good defines the threshold distinguishing justified from "antisocial" punishment. The methods integrate designs from previous, well-known experiments (Deck & Jahedi, 2015; Fehr & Gachter, 2000) involving human players engaged in cooperative games. As their major findings, the authors report that heavier cognitive loads increase players' propensities to apply punishment while significantly reducing the precision with which it is applied. This additional punishment disproportionately increases "false positives" (cooperators) while apparently leaving the punishment levels of true free riders relatively unaffected. The authors thus conclude that cognitive loads can severely undermine the efficacy of punitive action in public goods scenarios. This change in punishment behavior occurs within the context of reduced overall cooperation due to increased cognitive load, which has been reported previously.
I wish to commend the authors for writing the article (especially the Introduction and Methods) concisely and clearly, and in accessible terms. The overall concept extends and combines previous results in a relatively straightforward manner, but this combination seems novel and interesting, and its importance is made clear to readers. My comments that follow range from very minor issues to some larger questions. Any critiques related to experimental design are instead intended to prompt the authors to consider engaging with the issue at hand more directly in their exposition and discussions, rather than to "re-do" anything. I understand that it is not feasible to repeat the experiments, nor do I wish to suggest that the authors should do so. However, by addressing some of the issues raised more directly in text, I believe that the article's arguments could be strengthened and clarified.
____________________________
1) Lines 45-46 (and beyond):
The use of an arbitrary memorization task to control "cognitive load" challenged my own assumptions coming into the article. I have no doubts that the memorization task selected indeed increases cognitive load. The chosen task is not necessarily inappropriate, and has roots in the work of Deck & Jahedi (2015). However, when initially reading the paper, I had assumed that the "cognitive load" at hand would emerge in an "endogenous" way from the structure of the public goods game itself, rather than arbitrary external factors. By "endogenous", I mean the added cognitive loads that would be encountered when, for instance, engaging in games with greater numbers of players (who could even be distinguished punished separately), or in games where players would interact with the same players over time and have some incentive to remember those players' past behaviors. I expect that in cases where the cognitive loads came from the structure of the game itself, players might handle things differently as they consider the "costs" associated with managing these cognitive loads and balancing them with other factors. Cases like these lie obviously beyond the scope of the current study, and may in fact not be of interest to the authors at all. But in any case, I think it would be helpful if the authors qualified their chosen scope by (1) more clearly by explicitly acknowledging the potential for different kinds of cognitive loads, with more or less direct relationships to the actual decision at hand, and (2) justifying the article's focus on the arbitrary, "external" cognitive load represented by a memorization task. I think that by addressing this distinction directly at this point in the text, the nature of the cognitive load at hand could be made clearer to readers. Alternatively, at the discretion of the authors, the issue could perhaps be addressed in the Discussion.
Related to this, my understanding is that a single number memorization spanned the entire iterated experiment. Was there any reason the task was not repeated between each round with new numbers, or why individuals were not prompted to recall the number after each round? What would be the implications if they were? (If a task were repeated after each round, for example, changes in performance over time could be tracked). Some more explicit justifications for experimental design details like these, or alternatively some mention of alternative possibilities could be made in the Discussion/Conclusion section.
2) Line 66: "We" --> "we"
3) Line 67: I would recommend introducing a reference to Table 1 earlier, perhaps here for example: "...and the presence of punishment (with or without; see Table 1)"
4) Line 72-... :
At appropriate points in these Methods paragraphs, references should definitely be made to the sub-sections of the Appendices containing the corresponding text was presented to players in that group. It currently seems that the Appendices appear out of nowhere, un-anchored to any reference or explanation within the body of the text. I would go even further and more explicitly clarify here within the main Methods section just what aspects of the game were actually explained to subjects prior to playing, rather than relegating this only to the texts of the Appendix subsections (for example). When and in what form(s) do players receive information about their peers' behavior?
Related to this, it seems the Appendices should be introduced with some appropriate contextualization, e.g., "The following text was presented to subjects in the [...] treatment group prior to playing the game: ...". I don't want to be presumptuous about the original language of the text, but if the text here is an English translation of that which the subjects actually saw, this should be noted (subject to author and Editor discretion, it may even be appropriate to include the original untranslated text, if any).
5) Line 74: "participants" --> "Participants"
6) Lines 78-79 (Eq. 1): As a matter of taste, I find "1.6*(gi+gj)/2" more instructive than simplifying to ".8". Also, is there any particular justification or background for setting the multiplicative "return on investments" factor to 1.6? Also, here or in the Discussion/Conclusion, you might discuss/explain/speculate about how you expect results would (or wouldn't) be affected if this parameter were adjusted.
7) Lines 83-85: Same issue as above regarding the punishment amplification factor of 3. Why this value, and what if it were reduced or increased?
8) Line 92:
What are the reasons for, and implications of, the choice to randomly re-pair individuals with new partners after each round? Here or in the Discussion/Conclusion, it might be helpful to address the implications/consequences, if any, of this versus other alternative protocol, such as consecutive repeated pairings with the same opponents. On the surface, at least, it might seem to readers that the role of trust and punishment could be different if players encountered the same partners repeatedly (and were made aware of this). Does that matter here? Why or why not? Some clarification would be helpful.
9) Line 122: "Figure 1". Is there any reason why this Figure doesn't include similar panels illustrating the temporal evolution of punishment levels (overall, "antisocial", and correct)?
10) Line 135 (also 158):
I would encourage the authors to think carefully about the use of the term "accelerate", which for some readers will carry the specific connotation of a rate of cooperation collapse that increases over time (through subsequent rounds). Are the "rates" in question the more or less constant slopes/gradients of the data in Fig. 1, and the "acceleration" referring to the differences between the slopes of different treatment group? There is a potential for confusion here that "acceleration" refers to change in slope of the curve OVER TIME, and that change occurs faster in one case than another. I think the authors intend to say that "cooperation collapse" occurs at a certain apparently fixed rate, and that this rate is faster in one case than another. As an alternative, the authors could rephrase this as e.g., "cognitive load leads to a significantly faster rate of cooperation collapse". Specific reference to regression slopes may(?) help to clarify this somehow.
11) Line 139-140 (Equation 3): It seems unusual to reference Equation 3 inline (just) before it is introduced.
12) Lines 140-142: "..is a variable that indicates..."... Does this mean for example that HCL=0 for the "low cognitive load" treatment group and HCL=1 for the "high cognitive load" group? If so, consider being explicit about that.
13) Line 148: "OLS" --> "Ordinary Least Squares" or "Ordinary Least Squares (OLS)"
14) Lines 164-165: The axes labels on Figure 2 are unreadably tiny. Consider resizing and/or rearranging the panels, or whatever measures are required to prevent this from being the case in the final setting of the paper.
15) Lines 165- : It seems there is some ambiguity about "average punishment rounds" and "average punishment tokens". Are the numbers of tokens averaged only over those rounds where a non-zero punishment was applied, or overall all rounds?
16) Lines 176-180: When presenting and interpreting results about the lack of increase in punishment of actual free riders, either here or elsewhere, it may be helpful to consider what was the fraction of actual free-riders correctly targeted for punishment both of the treatment groups (what one might refer to as the "recall" in a machine learning / classification context, as opposed to the "precision")? If basically all free riders were already being punished under Low Cognitive Load settings, and then High Cognitive Load increased the overall rate of punishment, then of course this could only increase the rate of "false positives" since "true positive" counts had saturated. Might this be what is happening? Or is something more subtle going on?
17) Line 186: In Table 4, the interpretation of the "Demographics" row full of "Yes" entries is unclear to me. What is this and what does it add? Also, the "Number of Subjects" and "Number of Observations" rows also appear to contain all identical entries. Given this, does it make sense to include these in a table, or would it be better to report these values inline in the text body or in the caption? ("All entries represent 130 subjects and 1,300 observations ...")
18) Line 194-194: If Xi "includes round numbers", why is there no "t" subscript? (It might not be immediately clear to readers). Also, why the prime symbol on "Xi" in Equation 5?
19) Line 224-228: This could be read as implying that the current study has explored issues of "cultural context"? Is there reason to believe that different cultural contexts have different overall levels of cognitive load? (If so, it would be interesting to mention).
20) Lines 229-... :
Here I am reminded of the rich literature (especially that associated with Elinor Ostrom and colleagues) that explores how successful self-management of "commons" and public goods often amounts to structuring collective activities and institutions in (oftentimes creative) ways so as to reduce the "costs" associated with surveillance/monitoring (to detect infractions). I think a more in-depth discussion of this issue would be heplful here to contextualize and qualify the study's results. It is mentioned in passing an Appendix via the text shown to subjects that players are informed about their peers' contributions following each game round (again, I found this important detail very easy to miss, and would prefer to see it explicitly mentioned in the main Methods text). But in real-world situations, there are often additional "costs" associated with collecting information about peers' behavior (i.e., monitoring/surveillance), even before this information can be acted upon through costly punishment. Furthermore, management of these costs is often THE determining factor in whether communities are able to self-manage despite the possibility of free riding (as covered extensively in e.g., Ostrom's "Governing the Commons"). The issues touch back upon the points in comment #1 above.
On a related note, many computational models of evolutionary PGGs on networked populations make big (often unstated/unexplored) assumptions about how agents make decisions (one example offhand is Santos et al., "Social diversity promotes the emergence of cooperation in public goods games." Nature 454.7201 (2008): 213-216.). For example, many of their results hinge entirely upon an assumption that agents imitate higher-fitness peers, and that agents simultaneously apply the same behavior (either 'cooperation' or 'free riding') in ALL of the multiple interactions/games in which they engage. I mention this here only because "cognitive load" could be a justification for the latter assumption: agents end up behaving the same in all interactions because they lack the cognitive capacity to simultaneously keep track of so many different games. This is potentially interesting because, in such networked PGG models, this mechanism can actually function to foster/spread cooperative behavior. An inability to handle "cognitive load" could end up having "positive" effects in some contexts. There is no need to explain all of that in detail, of course, but it could be helpful to acknowledge that possibility.
21) Lines 232-234: "different types of cognitive load". Could this include cognitive loads directly related to the game itself (see again comment #1)? How might this matter?
21) Lines 249 and beyond: Again, the Appendices suddenly seem to appear without reference or context. The Methods should have made reference to these at the appropriate points.
22) Lines 268-269 (Appendix): "We will randomly select one round from your ten rounds of experiments to become your final payoff." Similar to comment #8 above, what is the reasoning behind the use of a randomly selected round to represent the final payoff? Would players' incentives be altered in any consequential way by different choices of payoff value, say, the average across all trials? This may not be immediately clear to readers.
23) As a final thought: Were the subjects' success rates in the memorization task ever reported? (I might have missed it.) The article seems to imply a trade-off in allocating cognitive resources between the memorization task and the exercise of punishment. This could be investigated further using existing data: At the individual level, does greater success in the memorization task (say, number of digits successfully recalled) indeed coincide with harsher punishment behavior?
__________________________________________________________
Again, I commend the authors on the interesting manuscript, and look forward to reading the edited version.
Comments on the Quality of English LanguageThe language of the paper was very clear. See my comments for some minor typos.
Author Response
For detailed responses, please refer to the attachment.

Reviewer 2 Report
Comments and Suggestions for Authors
Referee Report for “The Impact of Cognitive Load on Cooperation and Antisocial Punishment: Insights from a Public Goods Game Experiment”
This article presents the results of an experiment that examines the effects of cognitive load on behaviour in public goods setting, in the presence of punishment opportunities. They find that cognitive load, by having participants memorize long numbers, tends to affect (negatively) the evolution of cooperation and the rates of punishment. I have several comments on this work, which I outline below.
First, they pointed at a significant gap in the relevant literature. However, they seem to miss one of the most relevant contributions to the literature, namely an interesting paper showing a theoretical connection between self-control/cognitive load and human cooperation: Dreber, A., Fudenberg, D., Levine, D. K., & Rand, D. G. (2014, December). Altruism and selfcontrol. In Available at Social Science Research Network (SSRN): http://ssrn. com/abstract (Vol. 2477454).
My most important concern is that this experiment has a strong exploratory flavour. What is the theoretical basis for these findings/relationships between self-control and cooperation in the pubic goods game? How many of these findings were pre-hypothesized? Of course, the main problem with this is that in the absence of clear hypotheses, it is difficult to interpret the statistical results.
There is a particularly tenuous relationship that I would like to focus on. What is the theoretical connection between depletion of cognitive resources needed for deliberative processes and anti-social punishment? What does the former have to do with the latter? It seems as a completely ad hoc explanation. For instance, consider the passage: “Second, we partially uncover the reasons behind the use of antisocial punishment. We find that when individuals face high cognitive costs and cannot sufficiently consider the costs and benefits of punishment, they tend to engage in antisocial punishment.” (Lines 56-59) This is too speculative, especially intended as a contribution to our understanding of antisocial punishment.
SMALLER COMMENTS:
From the methodological point of view: how was the sample size determined? On the basis of which type of analysis?
In addition, there are several small typos. For instance, Figures 1 and 2 are highly problematic. The numbers and text in the axes are impossible to read. A very careful proofread is needed.
In the instructions, using the words ‘punish your partner’ may induce certain types of behaviour. Why not use neutral language, as practiced in most of the experimental literature?
Comments on the Quality of English Language
The English is OK.
Author Response

(The authors gave the same response as above.)

Reviewer 3 Report
Comments and Suggestions for Authors
Please find the attached PDF.

Author Response

(The authors gave the same response as above.)

Round 2
Reviewer 3 Report
Comments and Suggestions for Authors
I like the revised version. My concerns have disappeared.
However, I still have a few minor comments as follows. I recommend that the authors carefully review the paper in detail to ensure there are no mistakes.
1, On page 1. "Furthermore, [Reference 20]..." should be "Furthermore, Fischbacher et al. [20]..."
2, On page 5: "Specifically, We..." should be "Specifically, we..."
3, On page 7: The symbols of ***, **, and * in the figures are too small. Especially, I could not distinguish ** and * in panels c and f.
